# Inhibitory Effect of Fisetin on α-Glucosidase Activity: Kinetic and Molecular Docking Studies

**DOI:** 10.3390/molecules26175306

**Published:** 2021-08-31

**Authors:** Beiyun Shen, Xinchen Shangguan, Zhongping Yin, Shaofu Wu, Qingfeng Zhang, Wenwen Peng, Jingen Li, Lu Zhang, Jiguang Chen

**Affiliations:** 1Jiangxi Key Laboratory of Natural Products and Functional Food, College of Food Science and Engineering, Jiangxi Agricultural University, Nanchang 330045, China; shenbeiyun1997@163.com (B.S.); shangguanxc_818@sina.com (X.S.); yinzp2008@sina.com (Z.Y.); wsf1969@163.com (S.W.); zhqf619@126.com (Q.Z.); enen928@163.com (J.L.); 2The Laboratory for Phytochemistry and Plant-Derived Pesticides, College of Agriculture, Jiangxi Agricultural University, Nanchang 330045, China; pengwenwen123@sina.com; 3Collaborative Innovation Center of Jiangxi Typical Trees Cultivation and Utilization, College of Forestry, Jiangxi Agricultural University, Nanchang 330045, China; zhanglu856@mail.jxau.edu.cn

**Keywords:** α-glucosidase inhibition, fisetin, diabetes, molecular docking

## Abstract

The inhibition of α-glucosidase is a clinical strategy for the treatment of type 2 diabetes mellitus (T2DM), and many natural plant ingredients have been reported to be effective in alleviating hyperglycemia by inhibiting α-glucosidase. In this study, the α-glucosidase inhibitory activity of fisetin extracted from *Cotinus coggygria* Scop. was evaluated in vitro. The results showed that fisetin exhibited strong inhibitory activity with an IC_50_ value of 4.099 × 10^−4^ mM. Enzyme kinetic analysis revealed that fisetin is a non-competitive inhibitor of α-glucosidase, with an inhibition constant value of 0.01065 ± 0.003255 mM. Moreover, fluorescence spectrometric measurements indicated the presence of only one binding site between fisetin and α-glucosidase, with a binding constant (lgKa) of 5.896 L·mol^−1^. Further molecular docking studies were performed to evaluate the interaction of fisetin with several residues close to the inactive site of α-glucosidase. These studies showed that the structure of the complex was maintained by Pi-Sigma and Pi-Pi stacked interactions. These findings illustrate that fisetin extracted from *Cotinus coggygria* Scop. is a promising therapeutic agent for the treatment of T2DM.

## 1. Introduction

Diabetes mellitus (DM), associated with carbohydrate metabolism disorders and characterized by high blood glucose levels, has become a severe global non-communicable disease [1]. With 463 million people around the world suffering from DM, it is one of the top ten causes of death globally. DM causes about 1.5 million deaths per year worldwide [1,2]. It is estimated that the total number of individuals with diabetes worldwide will increase to 700 million by 2045. [3]. Two major types of diabetes exist, namely, type 1 and type 2 diabetes mellitus (T1DM and T2DM). T1DM primarily occurs as a result of autoimmune-mediated destruction of insulin-producing beta cells in the pancreas, leading to a lack of insulin in the body. T2DM, which accounts for approximately 90% of all diabetes cases [4] results from insulin resistance and relatively insufficient insulin secretion, leading to hyperglycemia [5]. α-Glucosidase participates in glucose metabolism by catalyzing the decomposition of polysaccharides into glucose in the small intestine [6]. One of the critical methods to treat DM is to reduce postprandial hyperglycemia. The digestion and absorption of dietary carbohydrates were delayed by inhibiting the digestive enzyme α-glucosidase [7,8]. Various representative α-glucosidase inhibitors have been reported to treat T2DM, such as acarbose, miglitol, and voglibose. However, the continuous use of these drugs is often accompanied by undesirable side effects, such as flatulence and abdominal discomfort [5]. Moreover, drug tolerance gradually increases with the continuous use of these drugs. Consequently, there is an urgent need to develop new potent α-glucosidase inhibitors with fewer adverse effects for treating T2DM. In recent years, the identification of new, safe, and efficient natural plant-derived inhibitors that can replace existing inhibitors has become a research hotspot in this field [9].

An increasing number of studies have shown that plant-derived flavonoids are particularly important in our diet, as they decrease the risk of cardiovascular diseases, neurodegenerative diseases, and DM [10]. Fisetin (3,3′,4′,7-tetrahydroxyflavone, Figure 1) is an essential bioactive flavonoid found in *Cotinus coggygria* Scop., which is distributed in Asia and North America [11] as a resource plant with great utilization value. It has various pharmacological properties, such as antimicrobial, antioxidant, anti-inflammatory, hepatoprotective, anti-tumor, and immunostimulant properties [12,13,14]. Messaadia et al. demonstrated that fisetin exhibits the best antioxidant activity with a high free radical scavenging capacity, followed by catechin and apigenin [15]. Choi et al. [16] found that the flavonoid fisetin (FI) may be useful for ameliorating the deleterious effects of high-fat diet (HFD)-induced obesity and related metabolic complications, such as hyperlipidemia, nonalcoholic fatty liver disease (NAFLD), hepatic fibrosis, and insulin resistance (IR). Furthermore, fisetin has been reported to be a potential agent for the development of novel strategies for diabetes treatments that suppress oxidative stress under hyperglycemic conditions in human monocytes [17]. Madeswaran et al. [18] compared the α-amylase inhibitory activity of flavonoids using in silico docking studies and found that baicalin had the strongest inhibitory activity, followed by apigenin, fisetin, wogonin, and scopoletin. Because of the phenolic hydroxyl structure, the flavonoids can inhibit α-amylase and α-glucosidase activity, thus retarding the absorption of glucose [8]. Yue et al. [19] investigated the in vitro inhibitory activity of fisetin on α-glucosidase, determined its inhibition type, and further conducted a molecular docking analysis to demonstrate the inhibitory mechanism.

However, the inhibitory mechanism of fisetin on α-glucosidase has not been fully understood. In the present paper, the enzyme inhibition kinetics of fisetin on α-glucosidase were performed, and the binding site, binding constant, and quenching mechanism were determined by fluorescence spectrum scanning, in addition, homology modeling and molecular docking were also carried out to elucidate the binding pattern, binding forces, and binding sites. Our study provided a theoretical basis for the further development and utilization of fisetin.

## 2. Materials and Methods

### 2.1. Chemicals and Reagents

α-Glucosidase and dimethyl sulfoxide (DMSO) were purchased from Shanghai Trading Co., Ltd. (Shanghai, China). Fisetin (98%) was obtained from Nanjing Yuan Plant Biology Co., Ltd. (Nanjing, China), and p-nitrophenyl-α-d-glucopyranoside (pNPG) was procured from Shanghai Huacheng Industrial Development Co., Ltd. (Shanghai, China). Phosphate buffer was purchased from Xiamen Haibiao Technology Co., Ltd. (Xiamen, China). Enzyme inhibition studies were performed using a 96-well microplate spectrophotometer (Thermo Scientific Multiskan FC, Waltham, MA, USA).

### 2.2. α-Glucosidase Inhibition Assay

The α-glucosidase inhibitory activity was determined as previously described by Chen et al. [20]. A total of 160 μL phosphate buffer (pH 6.8), 10 μL of α-glucosidase, and 10 μL of the test sample dissolved in DMSO were added to 96-well plates, and then the mixtures were incubated at 37 °C in a microplate reader for 20 min, oscillating for 10 s every 2 min. Meanwhile, *p*-nitrophenyl-β-glucopyranoside (pNPG, 10 mM) was preincubated at 37 °C in a water bath. Finally, 20 μL pNPG was added to the mixture to measure the change in absorbance at 405 nm with an interval of 2 min. Because the number of products produced has a linear relationship with the light absorption value, the reaction rate of the enzyme can be obtained by measuring the change in the light absorption value per unit time in a specific period. All experiments were performed in triplicate (mean ± SEM, *n* = 3). Percent inhibition was calculated using the following equation:(1)$$\alpha -\mathrm{glucosidase}\;\mathrm{activity}(\%)=\frac{A_{sample}}{A_{control}}\times 100\%$$
where *A* is the absorbance growth slope, *A_control_* is the enzyme activity without the inhibitor, and *A_sample_* is the enzyme activity with the inhibitor. The data obtained were used to determine the inhibitor concentration that inhibited 50% of the enzyme activity (IC_50_) using the GraphPad Prism 8.0.2 software (GraphPad Inc., La Jolla, CA, USA).

### 2.3. α-Glucosidase Kinetic Assay

The mode of inhibition of α-glucosidase activity by fisetin was determined using a Lineweaver–Burk plot, as previously described [21], in the presence and absence of the inhibitor. Different concentrations of pNPG (0.6–4 mM) were used as substrates in this assay.
(2)$$V=\frac{V_{m\;}\;S}{\;K_{m}\left(1+\frac{I}{K_{i}}\right)+S\left(1+\frac{I}{\alpha K_{i}}\right)\;}$$

Kinetic experiment data were analyzed using nonlinear regression, the GraphPad Prism 8.0.2 software (GraphPad Inc., La Jolla, CA, USA), and the Michaelis-Menten equation (where *V* is the initial velocity in the absence and presence of the inhibitor, respectively, *V_m_* is the maximum enzyme velocity, *S* and *I* are the concentrations of substrate and inhibitor, respectively, *K_m_* is the Michaelis-Menten constant, and *K_i_* and α*K_i_* are the competitive inhibition constant and the uncompetitive inhibition constant, respectively).

### 2.4. Fluorescence Spectrophotometric Measurement

The interaction between α-glucosidase and fisetin was analyzed using a 907CRT spectrophotometer, following the method described by Jang Hoon Kim et al. [22]. The fluorescence intensity of 0.5 U/mL enzyme solution mixed with different concentrations of fisetin (0–0.035 mM) was measured at 280 nm. The emission wavelength was 280–450 nm, the slit wavelength was 10 nm, and the scanning speed was medium.

### 2.5. Homology Modeling and Molecular Docking

Although the crystallographic structure of α-glucosidase from *Saccharomyces cerevisiae* is unavailable, X-ray crystal structures of a few bacterial α-glucosidases have been reported [23]. We carried out homology modeling of α-glucosidase from Baker’s yeast using the procedure described by Lin et al. [24]. The sequence of MAL32 α-glucosidase from *S. cerevisiae* was obtained from the NCBI database ( http://www.ncbi.nlm.nih.gov/protein/) (accessed on 10 June 2020). The protein 3AJ7_A, which showed high sequence similarity (72.34%) with α-glucosidase using the SWISS-MODEL server (SWISS-MODEL (expasy.org) (accessed on 10 June 2020)), was selected as the template for homology modeling [25]. To explore the binding mode of fisetin, molecular docking simulations were performed to predict the binding mode and force between α-glucosidase and fisetin using data obtained from PubChem. Autodock Tools (http://autodock.scripps.edu/resourc es/adt) (accessed on 14 June 2020) were used to remove the water molecules and add Gasteiger charges and essential hydrogen atoms [26]. The conformation energy was minimized for molecular interaction analyses through ligand molecule optimization and protein cleaning. Vina (http://vina.scripps.edu) (accessed on 14 June 2020) [27] was used for the molecular docking of fisetin and α-glucosidase. In the process of docking, the grid box parameter values for α-glucosidase were 18 ×18 ×18 Å, centered at X = 30.722, Y = −8.886, and Z = 10.506. After docking, molecular visualization was conducted using Pymol (https://pymol.org/2/) (accessed on 16 June 2020) [28] and Discovery Studio 2016 [28].

## 3. Results and Discussion

### 3.1. Inhibitory Effect of Fisetin on α-Glucosidase

As depicted in Figure 2A, the inhibitory activity of fisetin against α-glucosidase was investigated using acarbose as a positive control. The inhibitory effect was significantly affected by the fisetin concentration. When the concentration of fisetin increased, the relative enzyme activities rapidly decreased. The α-glucosidase inhibition rate of fisetin (2.0 × 10^−3^ mM) was 81.90%, while that of acarbose (2.0 × 10^−3^ mM) was 3.81%. The results showed that the IC_50_ values of fisetin and acarbose were 4.099 × 10^−4^ and 1.498 mM, respectively. Lower IC_50_ values indicated higher inhibition. Fisetin showed a stronger inhibitory effect on α-glucosidase than acarbose. Thus, it is speculated that fisetin is a potential α-glucosidase inhibitor. According to Medina-Pérez et al. [8], flavonoids can bind to biological polymer enzymes and inhibit α-glucosidase. Xu et al. [29] reported that the flavonoids myricetin and quercetin could have an inhibitory effect on α-glucosidase. Structurally, fisetin contains four phenolic hydroxyl groups, which can easily interact with residues at the enzyme active site, thereby inhibiting α-glucosidase activity [30]. ALTamimi et al. confirmed that fisetin protects against streptozotocin-induced diabetic cardiomyopathy in rats through hypoglycemic and insulin-releasing/sensitizing effects [31]. In summary, as a natural active substance that is relatively safe and easy to obtain [11], fisetin has a promising clinical application in inhibiting α-glucosidase.

### 3.2. Enzymatic Kinetic Assay

To study the mechanism underlying the inhibition of α-glucosidase by fisetin, an enzymatic kinetic study was carried out. The results in Figure 3A show that the slope increased, indicating that V_max_ decreased. All the data lines almost intersected with the X-axis at one point, namely K_m_ (0.1662 mM), and remained unchanged with the increase in the concentrations of fisetin compared to those of the samples without the inhibitor. The above analysis indicated that the inhibition type of fisetin on α-glucosidase was non-competitive inhibition. Similar determination results were reported by Yue et al. [19]. The K_i_ value was determined through the secondary re-plots of the Lineweaver-Burk plots, drawn by 1/V and 1/[S] (Figure 3B) [23,31]. The results confirmed that fisetin is a non-competitive inhibitor of α-glucosidase with a K_i_ of 0.01065 ± 0.003255 mM. In addition, the value of α was calculated to be 3.116, representing the effect of fisetin on the affinity of α-glucosidase for pNPG. In other words, fisetin tended to be more easily and firmly bound to the non-competitive domain instead of the active site [32]. Non-competitive inhibitors bind to the ABS instead of competing with pNPG for the OBS [33].

### 3.3. Fluorescence Quenching Studies of α-Glucosidase by Fisetin

To investigate the interaction between fisetin and α-glucosidase, fluorescence quenching experiments were performed. The fluorescence spectra shown in Figure 4A were obtained by varying the concentration of fisetin while keeping the α-glucosidase concentration unchanged. We found that with the increase in flavonoid concentration, the fluorescence intensity of fisetin was gradually quenched and the absorption peaks showed hypsochromic shifts.

The type of fluorescence quenching mechanism involved was analyzed using the Stern–Volmer equation [34,35]:
(3)$$\frac{F_{0}}{F}=K_{q}\tau_{0}\left[Q\right]+1$$
where *F_0_* and *F* represent the fluorescence intensities of α-glucosidase in the absence and presence of fisetin, respectively, [Q] is the concentration of fisetin, *K_q_* is the quenching rate constant of the biomolecule, and τ_0_ is the average lifetime of the fluorophore without the quencher (usually, τ_0_ = 1 × 10^−8^ S). According to Equation (3), *K_q_* was calculated to be 6.0501 × 10^12^ L·mol^−1^·s^−1^ (Figure 4B), which is much greater than 2.0 × 10^10^ L·mol^−1^·s^−1^. It was demonstrated that the fluorescence quenching process of fisetin on the enzyme involves a static quenching mechanism that forms a complex rather than a dynamic mechanism.

The binding constant (*K_a_*) and binding site (*n*) were determined according to the double logarithmic regression curve equation [36]:
(4)$$\mathrm{lg}\frac{\left(F_{0}-F\right)}{F}=lgK_{a}+nlg\left[Q\right]$$

The value of n was calculated to be 1.252, and *lgKa* was 5.896. The value of n was close to 1 [34], suggesting that fisetin was located at only one binding site on α-glucosidase. It was observed that the interaction of fisetin with α-glucosidase probably leads to the formation of a fisetin-α-glucosidase complex, resulting in a polarity variation of tryptophan (Trp) residues from hydrophilic to hydrophobic [37]. All the curves showed good linearity within the concentration range of 0–0.035 mM, suggesting that a single type of quenching existed in the fisetin-α-glucosidase static interaction. Similar results were reported by Şöhretoğlu et al., who showed the interaction between flavonoids and α-glucosidase through fluorescence quenching [26,38].

### 3.4. Homology Modeling and Molecular Docking Analysis of α-Glucosidase

To investigate the binding pattern between fisetin and α-glucosidase at the molecular level, a molecular docking analysis was conducted. Using the SWISS-MODEL server, we selected 3AJ7_A (identity: 72.34%) with the lowest discrete optimized protein energy (DOPE) score as the template [39,40]. A Ramachandran plot (Figure 5A) was used to show the soundness of the modeled α-glucosidase [41].

The theoretical binding mode between fisetin and α-glucosidase is shown in Figure 6. Fisetin did not compete with the pNPG substrate for the active site of α-glucosidase. Hence, fisetin is a non-competitive inhibitor of α-glucosidase, consistent with the results of the enzyme inhibition kinetic assay.

The amino acids in the 4 Å surrounding fisetin were analyzed in detail. As shown in Figure 7 and Figure 8A, fisetin is bound in a large hydrophobic pocket of α-glucosidase, surrounded by amino acid residues including ILE-415, PHE-420, ILE-416, PHE-232, PHE-311, PHE-430, PHE-310, ILE-315, and VAL-316. Furthermore, the fisetin phenyl ring showed a Pi-Sigma interaction at 3.51 Å with THR234 (Figure 8C). Moreover, a Pi-Pi stacked interaction formed between the phenyl ring of fisetin and PHE311, PHE420 of α-glucosidase, which played an essential role during the binding process [20].

Yue et al. [19] conducted a molecular docking analysis using 4J5T as a molecular crystal structure of α-glucosidase and found that fisetin could form the hydrogen bonds with TRP391, ASP392, ARG428, and ASP568 of α-glucosidase, which might result in a good inhibition activity on this enzyme. Unlike the above-mentioned investigation of Yue et al., 3AJ7_A was selected as the structure template for homology modeling in our study, and a grid number point (xyz 30.722, −8.886, 10.506) was chosen as the docking site because fisetin was a non-competitive inhibitor of α-glucosidase. In addition, besides the Pi-Sigma interaction, we also found the Pi-Pi stacked interaction between fisetin and enzyme. Consequently, we deduced that the Pi-Sigma interaction and Pi-Pi stacked interaction were the main driving forces of the combination between fisetin and α-glucosidase. Fisetin showed hydrophobic interactions with catalytic amino acid residues considered to be crucial in the catalytic mechanism, which further stabilized the receptor-ligand interactions and sequentially inhibited the catalytic activity [42]. The hydroxyl group of fisetin is a strong activating group that polarizes the molecule and enables it to interact with other residues multiple times [43]. Vaya et al. [44] found that adjacent hydroxyl groups on the B ring of flavonoids are probably conducive to the distribution of electron clouds that can donate hydrogen atoms to form hydrogen bonds with the active site residues of α-glucosidase more easily. Lin et al. [24] suggested that the Pi-Pi stacked interaction had a significant effect on the inhibitory activity of the inhibitor. According to the molecular docking results, the enzymatic inhibition mechanisms of fisetin were speculated to include a block of substrate entry by binding to the active pocket of α-glucosidase, conformation change of the active center of α-glucosidase, and decrease in enzyme activity. In summary, the above molecular docking analysis provides a rational explanation for the structure-activity relationship between fisetin and α-glucosidase, providing valuable information for further development of α-glucosidase inhibitors [23].

## 4. Conclusions

In the present study, the inhibitory effect of fisetin on α-glucosidase activity was investigated using enzyme inhibition, fluorescence quenching coupled with kinetic analysis, and molecular simulation. We found that fisetin has the potency to non-competitively inhibit α-glucosidase, with an IC_50_ value of 4.099 × 10^−4^ mM and K_i_ value of 0.01065 ± 0.003255 mM. The fluorescence quenching of α-glucosidase by fisetin was a static procedure. A single fisetin binding site was identified on α-glucosidase with a binding constant (*lgK_a_*) of 5.896. Comparison of the kinetic parameters between the IC_50_ and K_i_ slope showed that the inhibition tended to be non-competitive. Molecular docking was used to simulate binding interactions. The Pi-Sigma and Pi-Pi stacked interactions played a significant role in the inhibition ofα-glucosidase by fisetin. This study provides the foundation for the potential use of fisetin as an α-glucosidase inhibitor. However, we still need further in vivo studies on α-glucosidase to confirm our in vitro results.

## Figures and Tables

**Figure 1 molecules-26-05306-f001:**
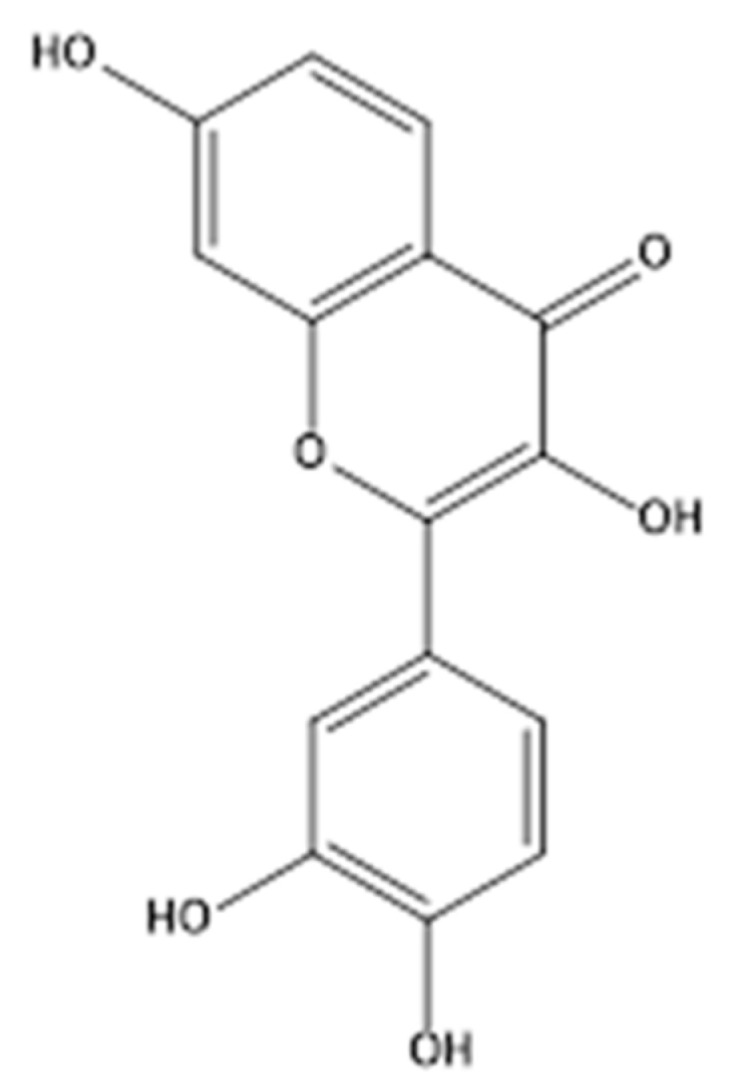
Molecular structure of fisetin.

**Figure 2 molecules-26-05306-f002:**
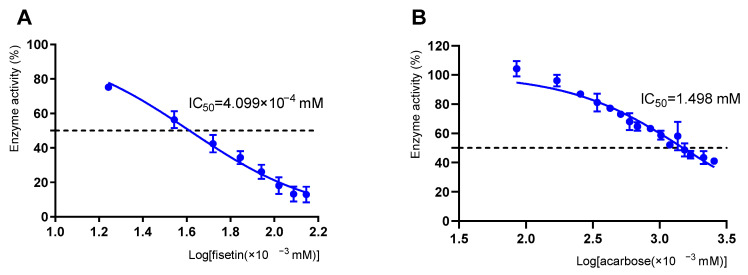
α-glucosidase inhibitor assay. (**A**) Inhibitory activity of fisetin against α-glucosidase. (**B**) Inhibitory activity of acarbose against α-glucosidase.

**Figure 3 molecules-26-05306-f003:**
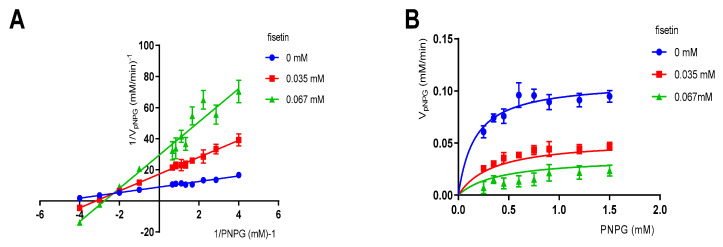
Enzymatic kinetic assay. (**A**) Lineweaver-Burke plot of the inhibition kinetics of α-glucosidase by fisetin. (**B**) Michaelis-Menten plot for fisetin in the presence of p-nitrophenyl-β-glucopyranoside (pNPG) at several concentrations for the determination of Ki.

**Figure 4 molecules-26-05306-f004:**
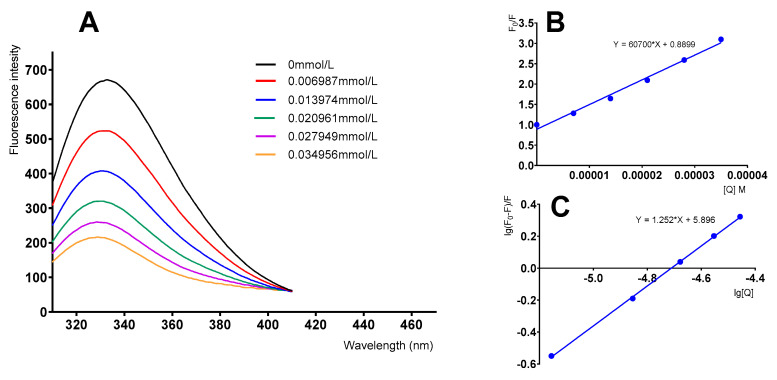
Fluorescence quenching studies of α-glucosidase by fisetin. (**A**) Fluorescence spectra of α-glucosidase in the presence of fisetin at different concentrations. (**B**) Stern-Volmer plots for α-glucosidase quenching by fisetin. (**C**) Plots of *lg(F_0_ − F)/F* versus *lg[Q]* for the fisetin-glucosidase system.

**Figure 5 molecules-26-05306-f005:**
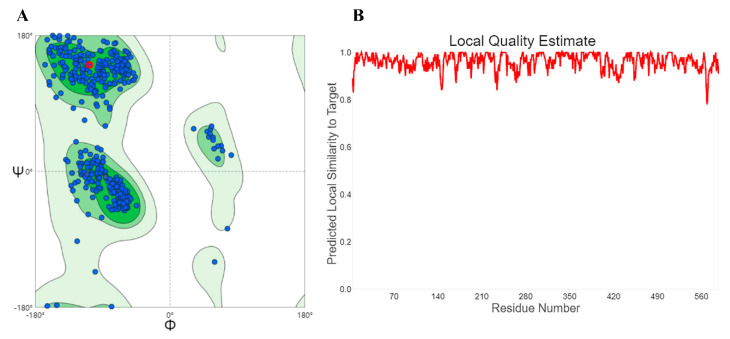
Homology modeling of α-glucosidase. (**A**) Ramachandran plot of the selected model protein. (**B**) The local quality estimate of the selected model protein.

**Figure 6 molecules-26-05306-f006:**
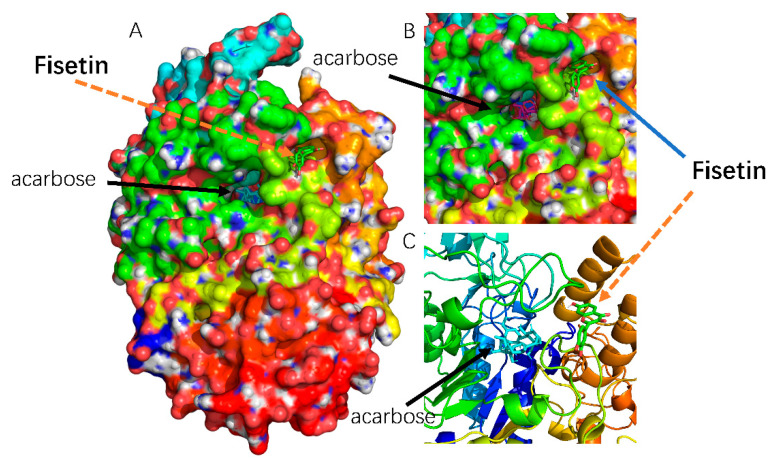
(**A**) Predicted binding sites of fisetin and acarbose docked with α-glucosidase. (**B**) The zoomed-in view of the binding locus on molecular surface. (**C**) The zoomed-in view of the binding sites docked with α-glucosidase drawn by cartoon image.

**Figure 7 molecules-26-05306-f007:**
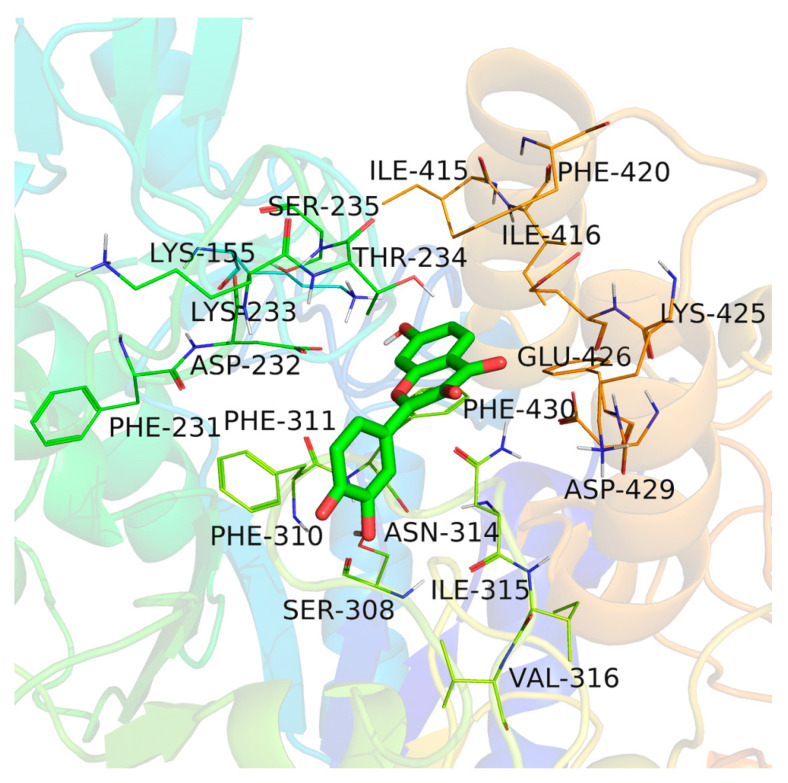
3-D interaction between fisetin and α-glucosidase.

**Figure 8 molecules-26-05306-f008:**
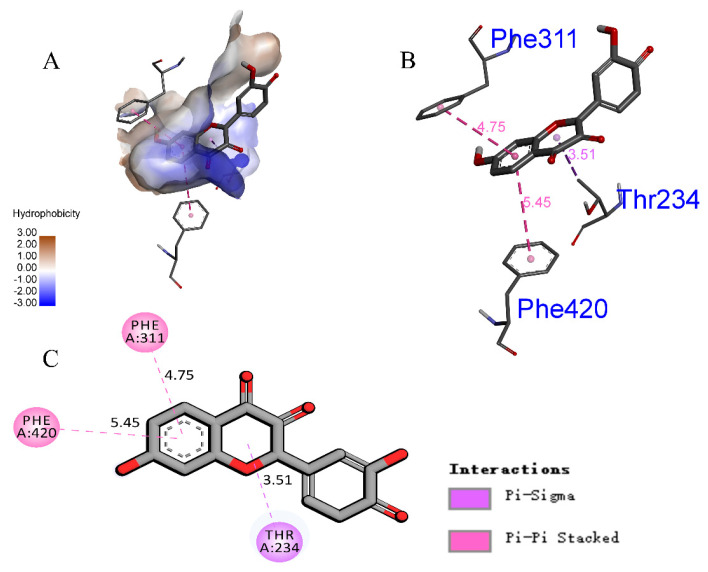
Molecular docking analysis of α-glucosidase. (**A**) Hydrophobicity of fisetin docked with α-glucosidase. (**B**) 3-D interaction between fisetin and α-glucosidase. (**C**) 2-D interactions of fisetin in the binding pocket of the developed homology model of α-glucosidase.

## Data Availability

Data are contained within the manuscript.

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
