# Peer review of "Inhibitory Effect of Fisetin on α-Glucosidase Activity: Kinetic and Molecular Docking Studies"

_molecules, 2021, doi:10.3390/molecules26175306_

Round 1
Reviewer 1 Report
Shen et al evaluated the inhibitory effect of fisetin on α-glucosidase in vitro. They further carried out enzyme kinetics and fluorescence quenching experiments and established that fisetin is a non-competitive inhibitor of α-glucosidase, with fisetin binding solely in one hydrophobic pocket in the enzyme. They also used molecular modelling to analyze binding interactions with a homology model.
Overall, I find the work interesting and suitable for publication after minor corrections (see more comments on the reviewed manuscript).
line 3, edit title accordingly
line 5, in authors' affilations, ''b'' should come first and ''c'' follows
intalicize ''in vitro'' throughout
line 37, State the estimated number of people currently having diabetes and the number dying each year as a result.
line 54, Support your claim with a reference or take it out. Note, anticancer agents have been obtained from plants, meaning some plant metabolites can be toxic.
Line 58-59, Rewrite the sentence. It is difficult to tell if you are talking about Fisetin or the plant.
line 73, Extrapolating α-amylase activity to α-glucosidase activity. Why the assumption? state any literature to justify this.
Line 155-156, ''the inhibitory effect of α-glucosidase is closely related to flavonoids'', this sentence is vague, rewrite or take it out.
Line 161, ''In summary, as a natural active substance that is relatively safe and easy to obtain'' include a reference or generated data to justify this or take it out completely.
In the conclusion, take the first two sentences out. They do not form part of this work.
Line 273, '' anti-a-glucosidase'', should be modified to read '' alpha-glucosidase''

Reviewer 2 Report
This paper is aimed to explore the in vitro α-glucosidase inhibitory activity of fisetin extracted from Cotinus coggygria Scop. The manuscript is well written and scientifically sound.
Some recommendations to improve the quality of the article follow:
1) Please use type 1 and type 2 Diabetes (in Arabic numbers) instead of type I and type II (in roman number, as per the old nomenclature).
2) Lines 36, 37, 38 (introduction) address on epidemiology. I suggest adapting the data (and references) to the 2019 International Diabetes Federation (IDF) Atlas – 9th Edition.
3) Line 45: it is said: “One of the critical methods to alleviate DM is to delay the digestion and absorption of dietary carbohydrates…”. What should be understood by ‘alleviate’? To reduce glycemia? To control glycemia? To prevent complications? I suggest modifying the wording to reduce imprecision.
4) Line 261: it is stated: “However, the side effects of acarbose are a limitation of this treatment. Therefore, the development of α-glucosidase inhibitors with few side effects and reliable efficacy based on traditional Chinese medicine is an important future research direction”. This paragraph suggests a potential avoidance (or risk reduction) of some adverse events. I suggest reconsidering the phrase to avoid an overinterpretation of the results. It is a legitimate intention to develop new alpha glucosidase inhibitors to increase efficacy and safety, but to our understanding, paper objectives, methodology and results do not permit to extrapolate great data about safety or efficacy.
5) Minor changes: please delete “Firstname Lastname 1, First name Lastname 2 and Firstname Lastname 2” from the manuscript title.
Reviewer 3 Report
The work from Shen et al. reported the characterization of the α-glucosidase inhibitory activity of Fisetin, as potential anti-diabetes agent. Although the context and the methodologies are quite well described, the aims of the work are too generic and superficially presented. Moreover, this work does not possess a strong innovative character, as inhibition and binding studies of fisetin with α-glucosidase have been recently reported and described in a paper the author missed to quote (Yue et al. Antioxidant and α-Glucosidase Inhibitory Activities of Fisetin. Natural Product Communications Vol. 13 (11) 2018). Although the present work get more insight into the binding mode of fisetin, some issues must be addressed before it can be accepted for publication. The results and discuss section should be also improved.
Line 54: “the biological toxicity of plant-derived medicines is low”. This sentence is too generic and incorrect as it is written. Their biological activities (and toxicity) are highly dependent on the species, components and so on. Please, express this concept more clearly and precisely.
Line 74: “However, the mechanisms of action of fisetin and α-glucosidase have not been scientifically investigated.” Again, this sentence is too generic and meaningless, also in consideration that other studies (see ref) already started to investigate more in depth the mechanism of action of fisetin.
Line 274: “We still need further in vitro studies on α-glucosidase to confirm our in vitro results”. This is a generic statement, what kind of studies do you want to perform? The authors should be more detailed about the planned studies.
Some other minor revisions need to be done:
Title: please correct “first name” which is in the title
Abstract: line 21 : specify the unit of the binding constant
Eq. 1: correct “control”
Line 161: please check the references, Altamimi et al. is ref 32.
Figure 8: Improve the quality
